# Innate Immunity in Children and the Role of ACE2 Expression in SARS-CoV-2 Infection

**Mario Dioguardi** [1,*] , **Angela Pia Cazzolla** [1] , **Claudia Arena** [1] , **Diego Sovereto** [1] , **Giorgia Apollonia Caloro** [2] ,
**Antonio Dioguardi** [3] , **Vito Crincoli** [4] , **Luigi Laino** [5] , **Giuseppe Troiano** [1] **and Lorenzo Lo Muzio** [1]

1   Department of Clinical and Experimental Medicine, University of Foggia, Via Rovelli 50, 71122 Foggia, Italy;
    elicio@inwind.it (A.P.C.); claudia.arena@unifg.it (C.A.); diego_sovereto.546709@unifg.it (D.S.);
    giuseppe.troiano@unifg.it (G.T.); lorenzo.lomuzio@unifg.it (L.L.M.)
2   Unità Operativa Nefrologia e Dialisi, Presidio Ospedaliero Scorrano, ASL (Azienda Sanitaria Locale) Lecce,
    Via Giuseppina Delli Ponti, 73020 Scorrano, Italy; giorgiacaloro1983@hotmail.it
3   U.S.C.A. "Unità Speciali di Continuità Assistenziale" Troia 2, ASL "Azienda Sanitaria Locale" Foggia
    Contrada Fontanelle, 71029 Troia, Italy; antoniodioguardi@gmail.com
4   Department of Basic Medical Sciences, Neurosciences and Sensory Organs, Division of Complex Operating
    Unit of Dentistry, "Aldo Moro" University of Bari, Piazza G. Cesare 11, 70124 Bari, Italy; vito.crincoli@uniba.it
5   Multidisciplinary Department of Medical-Surgical and Odontostomatological Specialties, University of
    Campania "Luigi Vanvitelli", 80121 Naples, Italy; luigi.laino@unicampania.it
*   Correspondence: mario.dioguardi@unifg.it

**Abstract:** COVID-19 (Coronavirus Disease 2019) is an emerging viral disease caused by the coronavirus SARS-CoV-2 (severe acute respiratory syndrome coronavirus 2), which leads to severe respiratory infections in humans. The first reports came in December 2019 from the city of Wuhan in the province of Hubei in China. It was immediately clear that children developed a milder disease than adults. The reasons for the milder course of the disease were attributed to several factors: innate immunity, difference in ACE2 (angiotensin-converting enzyme II) receptor expression, and previous infections with other common coronaviruses (CovH). This literature review aims to summarize aspects of innate immunity by focusing on the role of ACE2 expression and viral infections in children in modulating the antibody response to SARS-CoV-2 infection. This review was conducted using the Preferred Reporting Items for Systematic Reviews and Meta-Analyses (PRISMA) guidelines. Articles deemed potentially eligible were considered, including those dealing with COVID-19 in children and providing more up-to-date and significant data in terms of epidemiology, prognosis, course, and symptoms, focusing on the etiopathogenesis of SARS-CoV-2 disease in children. The bibliographic search was conducted using the search engines PubMed and Scopus. The following search terms were entered in PubMed and Scopus: COVID-19 AND ACE2 AND Children; COVID-19 AND Immunity innate AND children. The search identified 857 records, and 18 studies were applicable based on inclusion and exclusion criteria that addressed the issues of COVID-19 concerning the role of ACE2 expression in children. The scientific literature agrees that children develop milder COVID-19 disease than adults. Milder symptomatology could be attributed to innate immunity or previous CovH virus infections, while it is not yet fully understood how the differential expression of ACE2 in children could contribute to milder disease.

**Keywords:** COVID-19; SARS-CoV-2; coronavirus; ACE-2; adolescents; children

## 1. Introduction

The main coronaviruses involved in outbreaks in the last 20 years were SARS-CoV in 2003 [1], MERS (Middle East Respiratory Syndrome) in 2012 [2,3], and SARS-CoV-2 in 2019, which causes coronavirus disease 19 (COVID-19) [4].

COVID-19 is an emerging viral disease caused by the coronavirus SARS-CoV-2, which leads to severe respiratory infections in humans. The first reports came in December 2019 from Wuhan, Hubei, China [5].

Interpretation of the data from the beginning of the pandemic, and partially confirmed later, revealed that the main risk factors for admission to intensive care with a fatal outcome were advanced age [6], male sex [7], smoking [8], obesity [9,10], hypertension [11], diabetes mellitus [12–15], lung disease [16], cancer [17,18], and cardiovascular disease [19–21].

Animal studies on SARS-CoV infection have shown an age-dependent innate immune response, with non-human primates having more intense reactions with age than young adults [22].

Similarly, it was initially reported that children developed a milder disease than adults [21,23–26]. This could be due to a collective loss of immune protection due to aging, which leads to cellular and molecular dysregulation of the innate immune system. However, as the pandemic progressed, it became clear that children were less severely affected, while only 2.5% of laboratory-confirmed cases of SARS-CoV-2 in children were reported to develop severe disease associated with a cytokine storm similar to that of secondary hemophagocytic lymphohistiocytosis (HLH) [10,25].

Children can contract COVID-19 without gender differences and independently of age. In addition, clinical symptoms in children with COVID-19 are generally milder than those in adult patients, often without developing (severe) symptoms, while infants, in particular, are more vulnerable to SARS-CoV-2 infections.

According to Felsenstein and Hedrich, the reasons for mild manifestations in childhood could depend on cross-reactive antibodies and co-clearance with other viral infections, frequent contact with seasonal coronaviruses, and an increased expression of ACE2 in young people, which can facilitate virus infection but limit inflammation and reduce the risk of serious disease due to its involvement in anti-inflammatory signaling [27].

Additional potential age-related factors include recent vaccinations and associated heterologous immune responses and a more diverse memory T cell repertoire than in elderly populations.

These studies suggest that a lower severity of the pathology in children may depend on factors related to ACE2 expression and on a greater immunological diversification by T lymphocytes in the child induced by recent vaccinations and infections in the first years of life.

A significant cohort difference in the expression of ACE2 between COVID-19-positive adult and children patients emerged in some studies. For example, Zhang et al., who conducted a retrospective study on a cohort of 299 (173 children and 126 adults) COVID-19 patients, investigated the expression and distribution of ACE2 and lung progenitor cells and found that ACE2-positive cells and lung progenitor cells were decreased in the elderly, mainly in the lower pulmonary tract [28]. Additionally, Yonker et al., who conducted a cohort study of 192 children, found no correlation between ACE2 expression and viral load, suggesting that although higher ACE2 expression increased susceptibility to infections, once infected, children could carry high viral loads regardless of ACE2 expression levels; within the pediatric cohort, ACE2 expression increased with age [29].

In a study on the immune response of the nasal mucosa, Koch et al. reported that the expression of ACE2 and TMPRSS2 (transmembrane serine protease protein 2) was not linked with age and that the different immune responses to the virus determined the severity of disease, regardless of viral load [30].

From these studies, it could be hypothesized that children could have a high viral load during COVID-19 infection, regardless of the level of expression of ACE2 (which increases with age) and that it is not the viral load that determines the severity of the disease in children but the different immune response to the virus.

Therefore, it is clear that the differential expression of ACE2 cannot solely justify the pathogenicity of the disease, but all aspects of the innate immunity of children must be taken into consideration.

The reason that young patients develop serious consequences from COVID-19 remains uncertain and is a source of debate; this literature review aims to investigate and summarize aspects related to innate immunity by focusing on the role of the expression of ACE2 and

the role of viral infections in children in modulating the antibody response to SARS-CoV-2 infection.

The increase in cases of patients infected by COVID-19 and the presence of SARS-CoV-2 variants make it essential to acquire more information regarding the etiopathogenesis of the disease. Furthermore, the quantity of data provided and the number of publications in the last two years (2020–2021) concerning COVID-19 has reached a high number (130,000 articles). Therefore, our review further aims to summarize the aspects described above to provide researchers and doctors with the most up-to-date information within the scientific panorama; the reason for this review is to respond to the need for easily accessible scientific information on COVID-19 from the scientific community.

Researchers will find in this review all the most up-to-date information on the role of the differential expression of ACE2 in innate immunity in children.

## 2. Materials and Methods

This review was conducted using the Preferred Reporting Items for Systematic Reviews and Meta-Analyzes (PRISMA) guidelines [31]. After an initial screening phase, eligible studies were included in a qualitative analysis.

The studies considered were literature reviews, in vivo studies, in vitro studies, and related clinical studies that dealt with innate immunity in children against the SARS-CoV-2 virus, conducted in the last two years and published in English.

The articles considered potentially admissible were those dealing with COVID-19 in children that provided the most up-to-date and significant data regarding the epidemiology, prognosis, course, and symptoms of the disease, focusing on the etiopathogenesis in children. Specifically, studies included were those on the role of the ACE2 receptor and, more generally, those about previous immunity induced by previous infections in the early years of life. The inclusion and exclusion criteria are described in Table 1.

**Table 1.** Inclusion and exclusion criteria.

| Category | Exclusion Criteria | Inclusion Criteria |
|---|---|---|
| Publication Language | Not English | English |
| Study type | Reviews, systematic reviews [1], case reports, or case series | Clinical studies, in vitro studies, retrospective studies, prospective studies, cohort studies, clinical trials, and epidemiological studies |
| Data characteristics | Articles that did not report the number of patients/children, did not evaluate ACE2 expression or did not identify SARS-CoV-2 infections | Articles that reported data on the expression of ACE2 in children and the presence of the SARS-CoV-2 virus |
| Year of publication | Published before 2020 | Published in 2020–2021 |

[1] Systematic reviews and reviews were considered a source of bibliographic references.

Studies were identified using electronic databases using the "PubMed" and "Scopus" search engines. The search of electronic databases was conducted between 6 February 2021 and 10 March 2021; a partial update search of the literature was conducted on 15 March 2021.

The following search terms were entered into PubMed and Scopus: COVID-19 AND ACE2 AND Children, COVID-19 AND Immunity innate AND children. Filters for systematic reviews, reviews, and clinical studies were applied to find previous systematic reviews and avoid replicating previously considered results and hypotheses.

After the records were identified, overlaps were removed using Endnote X8 (Clarivate Analytics, London, UK). This research concerned the subsequent screening of records obtained and was carried out by two independent reviewers; uncertain positions were discussed with a third reviewer.

The screening included analysis of the title and abstract of studies to eliminate the records not related to the topics of the review. The potentially admissible articles were

subjected to full-text analysis to verify their use for qualitative analysis. A third reviewer resolved disagreements, and a fourth reviewer oversaw the study.

The two reviewers were M. D. and C. Q., while the third reviewer was G. T., dentists at the Department of Clinical and Experimental Medicine of the University of Foggia, Italy. The fourth reviewer, who oversaw the project, was L. Lo. M.

The entire selection and screening phase of the databases, the number of records, and the keywords used are reported in Table 2.

**Table 2.** Complete overview of the search methodology. Overlaps were removed using EndNote X8. Records identified by databases: 857; records selected for qualitative analysis: 18.

| Database/ Provider | Keywords, Search Details | Number of Records | Records after Removal of Overlapping Articles | Records after the Application of the Initial Eligibility Criteria | Articles Deemed Potentially Eligible | Articles Included in the Review that Discussed COVID-19 Issues in Children Regarding ACE2 Receptor Expression |
|---|---|---|---|---|---|---|
| **Pubmed** | Search: covid 19 AND Immunity innate AND children Sort by: Most Recent ("covid 19"[All Fields] OR "covid 19"[MeSH Terms] OR "covid 19 vaccines"[All Fields] OR "covid 19 vaccines"[MeSH Terms] OR "covid 19 serotherapy"[All Fields] OR "covid 19 serotherapy"[Supplementary Concept] OR "covid 19 nucleic acid testing"[All Fields] OR "covid 19 nucleic acid testing"[MeSH Terms] OR "covid 19 serological testing"[All Fields] OR "covid 19 serological testing"[MeSH Terms] OR "covid 19 testing"[All Fields] OR "covid 19 testing"[MeSH Terms] OR "sars cov 2"[All Fields] OR "sars cov 2"[MeSH Terms] OR "severe acute respiratory syndrome coronavirus 2"[All Fields] OR "ncov"[All Fields] OR "2019 ncov"[All Fields] OR (("coronavirus"[MeSH Terms] OR "coronavirus"[All Fields] OR "cov"[All Fields]) AND 2019/11/01:3000/12/31[Date—Publication])) AND ("immunity, innate"[MeSH Terms] OR ("immunity"[All Fields] AND "innate"[All Fields]) OR "innate immunity"[All Fields] OR ("immunity"[All Fields] AND "innate"[All Fields]) OR "immunity innate"[All Fields]) AND ("child"[MeSH Terms] OR "child"[All Fields] OR "children"[All Fields] OR "child s"[All Fields] OR "children s"[All Fields] OR "childrens"[All Fields] OR "childs"[All Fields]) Translations covid 19: ("COVID-19" OR "COVID-19"[MeSH Terms] OR "COVID-19 Vaccines" OR "COVID-19 Vaccines"[MeSH Terms] OR "COVID-19 serotherapy" OR "COVID-19 serotherapy"[Supplementary Concept] OR "COVID-19 Nucleic Acid Testing" OR "covid-19 nucleic acid testing"[MeSH Terms] OR "COVID-19 Serological Testing" OR "covid-19 serological testing"[MeSH Terms] OR "COVID-19 Testing" OR "covid-19 testing"[MeSH Terms] OR "SARS-CoV-2" OR "sars-cov-2"[MeSH Terms] OR "Severe Acute Respiratory Syndrome Coronavirus 2" OR "NCOV" OR "2019 NCOV" OR (("coronavirus"[MeSH Terms] OR "coronavirus" OR "COV") AND 2019/11/01[PDAT]: 3000/12/31[PDAT])) Immunity innate: "immunity, innate"[MeSH Terms] OR ("immunity"[All Fields] AND "innate"[All Fields]) OR "innate immunity"[All Fields] OR ("immunity"[All Fields] AND "innate"[All Fields]) OR "immunity, innate"[All Fields] children: "child"[MeSH Terms] OR "child"[All Fields] OR "children"[All Fields] OR "child's"[All Fields] OR "children's"[All Fields] OR "childrens"[All Fields] OR "childs"[All Fields] | 117 | | | | |

**Table 2.** *Cont.*

| Database/ Provider | Keywords, Search Details | Number of Records | Records after Removal of Overlapping Articles | Records after the Application of the Initial Eligibility Criteria | Articles Deemed Potentially Eligible | Articles Included in the Review that Discussed COVID-19 Issues in Children Regarding ACE2 Receptor Expression |
|---|---|---|---|---|---|---|
| Pubmed | Search: covid 19 AND ace2 AND Children Sort by: Most Recent ("covid 19"[All Fields] OR "covid 19"[MeSH Terms] OR "covid 19 vaccines"[All Fields] OR "covid 19 vaccines"[MeSH Terms] OR "covid 19 serotherapy"[All Fields] OR "covid 19 serotherapy"[Supplementary Concept] OR "covid 19 nucleic acid testing"[All Fields] OR "covid 19 nucleic acid testing"[MeSH Terms] OR "covid 19 serological testing"[All Fields] OR "covid 19 serological testing"[MeSH Terms] OR "covid 19 testing"[All Fields] OR "covid 19 testing"[MeSH Terms] OR "sars cov 2"[All Fields] OR "sars cov 2"[MeSH Terms] OR "severe acute respiratory syndrome coronavirus 2"[All Fields] OR "ncov"[All Fields] OR "2019 ncov"[All Fields] OR (("coronavirus"[MeSH Terms] OR "coronavirus"[All Fields] OR "cov"[All Fields]) AND 2019/11/01:3000/12/31[Date—Publication])) AND "ace2"[All Fields] AND ("child"[MeSH Terms] OR "child"[All Fields] OR "children"[All Fields] OR "child s"[All Fields] OR "children s"[All Fields] OR "childrens"[All Fields] OR "childs"[All Fields]) Translations covid 19: ("COVID-19" OR "COVID-19"[MeSH Terms] OR "COVID-19 Vaccines" OR "COVID-19 Vaccines"[MeSH Terms] OR "COVID-19 serotherapy" OR "COVID-19 serotherapy"[Supplementary Concept] OR "COVID-19 Nucleic Acid Testing" OR "covid-19 nucleic acid testing"[MeSH Terms] OR "COVID-19 Serological Testing" OR "covid-19 serological testing"[MeSH Terms] OR "COVID-19 Testing" OR "covid-19 testing"[MeSH Terms] OR "SARS-CoV-2" OR "sars-cov-2"[MeSH Terms] OR "Severe Acute Respiratory Syndrome Coronavirus 2" OR "NCOV" OR "2019 NCOV" OR (("coronavirus"[MeSH Terms] OR "coronavirus" OR "COV") AND 2019/11/01[PDAT]: 3000/12/31[PDAT])) Children: "child"[MeSH Terms] OR "child"[All Fields] OR "children"[All Fields] OR "child's"[All Fields] OR "children's"[All Fields] OR "childrens"[All Fields] OR "childs"[All Fields] | 273 | | | | |

**Table 2.** *Cont.*

| Database/ Provider | Keywords, Search Details | Number of Records | Records after Removal of Overlapping Articles | Records after the Application of the Initial Eligibility Criteria | Articles Deemed Potentially Eligible | Articles Included in the Review that Discussed COVID-19 Issues in Children Regarding ACE2 Receptor Expression |
|---|---|---|---|---|---|---|
| **Pubmed** | Search: covid 19 AND ace2 AND Pediatric Sort by: Most Recent ("covid 19"[All Fields] OR "covid 19"[MeSH Terms] OR "covid 19 vaccines"[All Fields] OR "covid 19 vaccines"[MeSH Terms] OR "covid 19 serotherapy"[All Fields] OR "covid 19 serotherapy"[Supplementary Concept] OR "covid 19 nucleic acid testing"[All Fields] OR "covid 19 nucleic acid testing"[MeSH Terms] OR "covid 19 serological testing"[All Fields] OR "covid 19 serological testing"[MeSH Terms] OR "covid 19 testing"[All Fields] OR "covid 19 testing"[MeSH Terms] OR "sars cov 2"[All Fields] OR "sars cov 2"[MeSH Terms] OR "severe acute respiratory syndrome coronavirus 2"[All Fields] OR "ncov"[All Fields] OR "2019 ncov"[All Fields] OR (("coronavirus"[MeSH Terms] OR "coronavirus"[All Fields] OR "cov"[All Fields]) AND 2019/11/01:3000/12/31[Date—Publication])) AND "ace2"[All Fields] AND ("paediatrics"[All Fields] OR "pediatrics"[MeSH Terms] OR "pediatrics"[All Fields] OR "paediatric"[All Fields] OR "pediatric"[All Fields]) Translations covid 19: ("COVID-19" OR "COVID-19"[MeSH Terms] OR "COVID-19 Vaccines" OR "COVID-19 Vaccines"[MeSH Terms] OR "COVID-19 serotherapy" OR "COVID-19 serotherapy"[Supplementary Concept] OR "COVID-19 Nucleic Acid Testing" OR "covid-19 nucleic acid testing"[MeSH Terms] OR "COVID-19 Serological Testing" OR "covid-19 serological testing"[MeSH Terms] OR "COVID-19 Testing" OR "covid-19 testing"[MeSH Terms] OR "SARS-CoV-2" OR "sars-cov-2"[MeSH Terms] OR "Severe Acute Respiratory Syndrome Coronavirus 2" OR "NCOV" OR "2019 NCOV" OR (("coronavirus"[MeSH Terms] OR "coronavirus" OR "COV") AND 2019/11/01[PDAT]: 3000/12/31[PDAT])) Pediatric: "paediatrics"[All Fields] OR "pediatrics"[MeSH Terms] OR "pediatrics"[All Fields] OR "paediatric"[All Fields] OR "pediatric"[All Fields] | 249 | | | | |
| **Scopus** | TITLE-ABS-KEY (covid 19 AND immunity AND innate AND children) | 94 | | | | |
| **Scopus** | TITLE-ABS-KEY (covid 19 AND ace2 AND children) | 95 | | | | |
| **Scopus** | TITLE-ABS-KEY (covid 19 AND ace2 AND pediatric) | 29 | | | | |
| **Total** | | 857 | 556 | 204 | 35 | 18 |

## 3. Results

From PubMed and Scopus database searches, 857 records were identified. EndNote X8 was used to remove overlaps, resulting in 556 records. After elimination of reviews, case reports, and articles not in English, 204 registrations were obtained. Then, following elimination of the articles that had little relevance to the themes of the review, 35 studies were obtained. In total, 18 studies were included in the review that dealt with COVID-19 and the role of ACE2 expression in children (Figure 1).

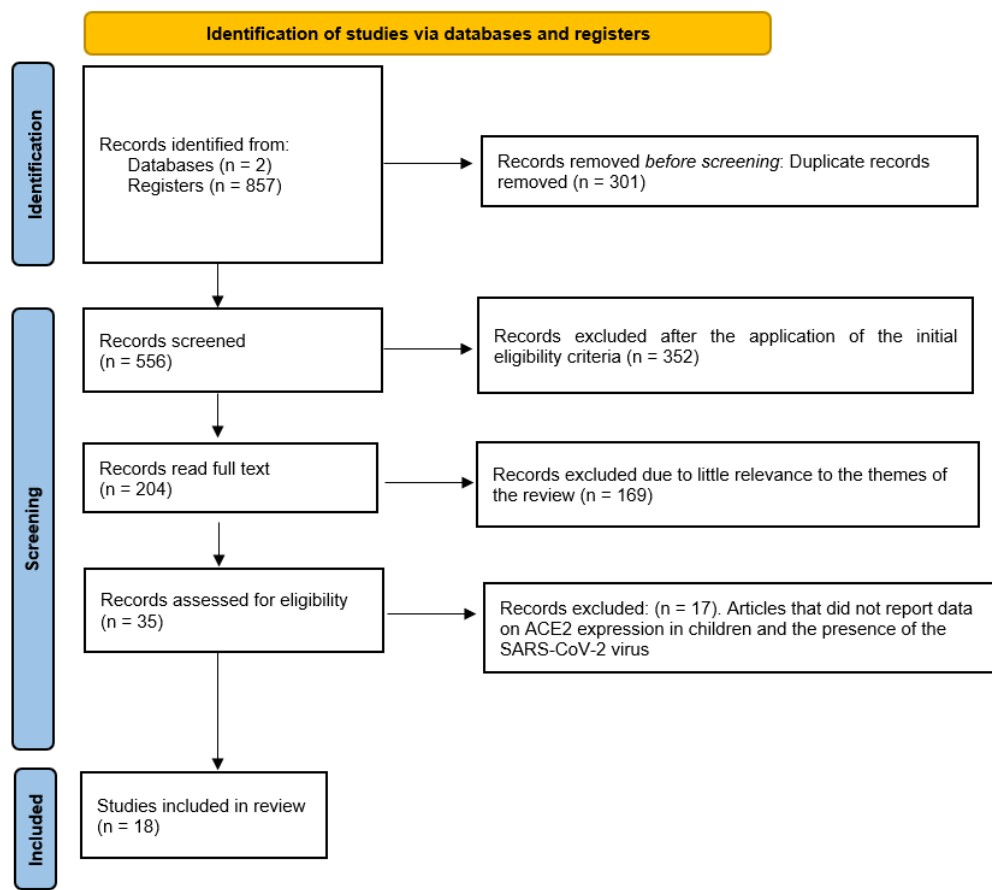

**Figure 1.** Flow chart of the different phases of the systematic review.

The studies included for the qualitative analysis were as follows: Scagnolari et al. [32], Sajuthi et al. [33], Somekh et al. [34], Bunyavanich et al. [35], Swärd et al., [36], Taglauer et al. [37], Vuille-Dit-Bille et al. [38], Zhang et al. [28], Galván-Peña et al. [39], Koch et al. [30], Ortiz Bezara et al. [40], Sharif-Askari et al. [41], Schweitzer et al. [42], Pavel et al. [43], Yonker et al. [29], Inde et al. [44], Zhang et al. [45], and Heinonen et al. [46].

Extracted data included first author, publication date, literature references, number of patients, type of sample investigated, virus sought, the SARS-CoV-2 receptor investigated, and the extracted results/conclusions.

The extraction of the data and reporting modalities were based on the Cochrane Handbook for Systematic Reviews of Interventions, Chapter 7 (Selection of Studies and Data Collection), particularly from pages 156–182.

The results are summarized in Table 3.

**Table 3.** Primary information extracted from the articles concerning the role of ACE2 expression in children related to the pathogenesis of COVID-19.

| First Author and Date | Patients | Number | Age, D.S. | Sample Type | Virus | Expression Receptor SARS-CoV-2 | Main Study Conclusions |
|---|---|---|---|---|---|---|---|
| Scagnolari et al. 2021 [32] | Children | 59 | 1.21 ± 2.45 | Nasopharyngeal washings | 14 respiratory viruses and SARS-CoV-2 | In vivo gene expression: ACE2, furin, GUS (beta-glucuronidase gene), and ISG15 (IFN-Stimulated Genes) | IFN (interferon) only increased the truncated ACE2 isoform; this activation would not increase the risk of SARS-CoV-2 infection in the respiratory tract. |
| | Adults | 48 | 61.67 ± 16.91 | Nasopharyngeal swabs | | | |
| Sajuthi et al. 2020 [33] | Children | 695 | – | Nasal airway brushings | CoV species (OC43, JKU1, 229E, and NL63), rhinovirus species C (HRV-C), Influenza A, Influenza B, Orthopneumovirus, and metapneumovirus, Enterovirus, or parainfluenza | ACE2 and TMPRSS2 | The response of interferon to respiratory viruses highly upregulated the expression of ACE2. IL-13-mediated and virus infection effects on ACE2 expression at the protein level in the airway epithelium were also observed. |
| | Adult | 1 | – | Nasal airway epithelium | | | |
| Somekh et al. 2020 [34] | Children | 31 | 5–17 | – | SARS COV 2 | ACE2 expression | The correlation between the two sets of values (sensory impairment scores and relative ACE2 expression) was 0.95 ($p = 0.05$). |
| | Adults | 42 | +18 | | | | |
| Bunyavanich et al. 2020 [35] | Children | 45 | Aged < 10 years, | Nasal epithelium | – | ACE2 expression | The age-dependent expression of ACE2 in the nasal epithelium. |
| | | 185 | older children (10–17 years) | | | | |
| | Adults | 46 | young adults (18–24 years), | | | | |
| | | 29 | and adults (≥25 years) | | | | |
| Zhang et al. 2021 [45] | Children | 173 | 0–1 years ($n = 36$), 1–5 years ($n = 41$) and 5–15 years ($n = 96$) | Nasopharyngeal swabs or sputum, biopsy samples (9 in each age group) | SARS-CoV-2 | ACE2 | Infants (<1-year-old) with SARS-CoV-2 infection were more vulnerable to lung injury. |

**Table 3.** *Cont.*

| First Author and Date | Patients | Number | Age, D.S. | Sample Type | Virus | Expression Receptor SARS-CoV-2 | Main Study Conclusions |
|---|---|---|---|---|---|---|---|
| Yonker et al. 2020 [29] | Children | 192 | 10.2 ± 7.0 | nasopharyngeal and oropharyngeal swabs and blood specimens | SARS-CoV-2 | ACE2 | Initial findings showed that although a low expression of ACE2 in younger children (<10 years of age) likely corresponds to reduced infection rates, children of all ages, once infected, can carry high SARS-CoV-2 viral loads. |
| Pavel et al. 2021 [43] | Children | 19 healthy 29 atopic dermatitis | Healthy infants and toddlers (≤5 years old, mean age: 2.1; 52.6% female) | Serum | – | ACE2 and CTSL1 (Cathepsin L1) | Data showed significantly higher ACE2 protein expression in the serum of adults compared with infants and toddlers and in adult males compared with adult females. These data suggest the potential systemic role of ACE2 protein levels in the differential clinical manifestations among various patient populations. |
|  | Adults | 17 healthy 55 atopic dermatitis | Healthy adults (age range: 24–55, mean age: 41; 35.3% female) |  |  |  |  |
| Swärd et al. 2020 [36] | Children | Males and females: 824 | >18 | Serum | – | ACE2 | Subjects with a higher risk of severe COVID-19 had a higher sACE2 (adults > children and men > women). |
|  | Adults | Males and females: 241 | <18 |  |  |  |  |
| Taglauer et al. 2020 [37] | Maternal-fetal dyads | 15, COVID-19 positive | Maternal age (years): Mean (SD) 31.8 (5.5), gestational age at birth (weeks): Mean (SD) 38.1 (1.7) | Placental tissue | SARS-CoV2 | ACE2 and TMPRSS2 | CoV2 SP (spike protein) and ACE2 expression were coherently localized mainly within the placental villi of the outer syncytiotrophoblastic layer. |
|  |  | 10 contemporary COVID-19 negative controls | Maternal age (years): mean (SD) 30.1 (5.5); Gestational age at birth (weeks): Mean (SD) 39.3 (1.6) |  |  |  |  |

**Table 3.** *Cont.*

| First Author and Date | Patients | Number | Age, D.S. | Sample Type | Virus | Expression Receptor SARS-CoV-2 | Main Study Conclusions |
|---|---|---|---|---|---|---|---|
| Vuille-Dit-Bille et al. 2020 [38] | Adults | 43 healthy | 60 (49–66) | Duodenal tissue | – | ACE2 | Increased intestinal ACE2 mRNA expression in elderly patients may affect their susceptibility to developing intestinal symptoms. |
| Zhang et al. 2021 [28] | Children | 173 | 0–16 years | Nasopharyngeal swabs | SARS-CoV-2 | ACE2 | Compared to children, ACE2-positive cells generally decreased in the elderly. |
| | Adult | 126 | 16–80 | | | | |
| Galván-Peña et al. 2020 [39] | Adults | 57 | 20–80 | SARS-CoV-2 | SARS-CoV-2 | Tregs and FOXP3 | Different identification of Treg lymphocytes in COVID-19 patients, which could impact the pathogenicity of COVID-19. |
| Ortiz Bezara et al. 2020 [40] | Children Adult | 29 cases | 0.5–71 years | Tissues included nasal biopsies ($n = 3$), lung donors ($n = 29$), and autopsy tissues (control tissues such as small intestine and kidney) | – | ACE2 | The ACE2 protein was highest within regions of the sinonasal cavity and pulmonary alveoli. In the lung parenchyma, the ACE2 protein was found on the apical surface of a small subset of alveolar type II cells and colocalized with TMPRSS2, a cofactor for SARS-CoV-2 entry. The ACE2 protein did not increase with pulmonary risk factors for severe COVID-19. Additionally, the ACE2 protein was not reduced in children, a demographic with a lower incidence of severe COVID-19. |
| Sharif-Askari et al. 2020 [41] | Children | 4 datasets for children groups (healthy and asthmatics) | – | Blood, upper and lower respiratory tract tissue, and saliva | – | ACE2 and TMPRSS2 | The difference in COVID-19 severity between children and adults was, in part, attributed to the difference in ACE2 and TMPRSS2 airway tissue expression levels. |
| | Adults | 15 datasets for adults with different comorbidities | | | | | |

**Table 3.** *Cont.*

| First Author and Date | Patients | Number | Age, D.S. | Sample Type | Virus | Expression Receptor SARS-CoV-2 | Main Study Conclusions |
|---|---|---|---|---|---|---|---|
| Schweitzer et al. 2021 [42] | | 100 | 4 months to 75 years of age. | Human lung tissue specimens | – | ACE2 and TMPRSS2 | Human small airway epithelial cells from healthy patients were subsequently infected with the influenza A virus, leading to an amplification of ACE2, sheddase ADAM17 (TACE), and TMPRSS2 expression, which are involved in the penetration of SARS-CoV-2 into cells. |
| Inde et al. 2020 [44] | Children Adults | | 9–75 years | Lung tissue specimens ($n$ = 100) | – | ACE2 and TMRPRSS2 | ACE2 expression in distal lung epithelial cells generally increased with advancing age but exhibited extreme intraindividual and interindividual heterogeneity. ACE2 expression was also detected on neonatal airway epithelial cells and within the lung parenchyma. |
| Koch et al. 2021 [30] | Children | 7 healthy | | Curettage of nasal mucosa | SARS-CoV-2, respiratory syncytial virus (RSV), and influenza virus (IV) | ACE2 and TMPRSS2 | No difference in ACE2 or TMPRSS2 expression was observed between children and adults. No increase in ACE2 and TMPRSS2 expression was observed during SARS-CoV-2 or other active viral infections. |
| | | 36 SARS-CoV-2 | 1.9 (0.4–15.0) | | | | |
| | | 24 RSV | 0.33 (0.16–0.44) | | | | |
| | | 9 IV | 1.7 (1.4–7.0) | | | | |
| | Adults | 13 healthy | 37 (31–42) | | | | |
| | | 16 SARS-CoV-2 | 31.5 (24.0–38.5) | | | | |
| Heinonen et al. 2021 [46] | Children (newborns) | 17 term | Gestational age: 40 + 0 ± 0.9 weeks | Nasal epithelium | – | ACE2, (TMPRSS2), neuropilin 1 (NRP1), neuropilin 2, (NRP2), and insulin-like growth factor 1 receptor (IGF1R) | Both term and preterm newborns, compared with adults, had lower expression of SARSCoV-2 entry receptors in the nasal epithelium. |
| | | 11 preterm | 30.1 ± 1.8 weeks | | | | |
| | Adults | 10 | 30–60 | | | | |

Only five studies met the inclusion criteria (the clear and precise distinction between adult and child patients, SARS-CoV-2 virus was found in patients, looked for a difference in the expression of ACE2); Koch et al. [30], Zhang et al. [28], Yonker et al. [29], Somekh et al. [34], and Taglauer et al. [37]. Despite the other thirteen studies meeting the eligibility and inclusion criteria, they had a deficient or reductive argument on one of these three points.

Here, we present the results of the five studies that met the inclusion criteria. Somekh et al. [34] found a correlation between SARS-CoV-2 and ACE2 expression by comparing the data with those previously published by Bunyavanich et al. on another patient cohort [35]; moreover, their research focused more on sensory taste and olfactory variations induced by the infection in adult patients and children.

Koch et al. conducted a study in children and adults on the immune response of the nasal mucosa to three viruses, SARS-CoV-2, respiratory syncytial virus (RSV), and influenza virus (IV), in the nasal mucosa in children and adults. They reported that the expression of ACE2 and TMPRSS2 was not related to age and that the different immune responses determined the severity of the disease, regardless of viral load [30].

In a retrospective study on a cohort of 299 (173 children and 126 adults) COVID-19 patients, Zhang et al., investigated the expression and distribution of ACE2 and lung progenitor cells and found that ACE2-positive cells and lung progenitor cells were decreased in the elderly, occurring mainly in the lower pulmonary tract [28].

Yonker et al., who studied a cohort of 192 children, found that there was no correlation between ACE2 expression and viral load, suggesting that although increased ACE2 expression increased susceptibility to infections once infected, children could carry high viral loads regardless of the level of ACE2 expression and, within the pediatric cohort, ACE2 expression increased with age. Furthermore, low ACE2 expression in younger children (<10 years of age) may correspond to reduced infection rates [29].

Taglauer et al. conducted a maternal–fetal dyad study in patients with SARS-CoV-2, investigating the expression of ACE2 and TMPRSS2 in the placental villi of 15 pregnant patients; the study identified the spike protein of SARS-CoV-2 in placental villi in COVID-19-positive pregnancies, with and without evidence of fetal transmission [37].

No previously performed review of the systematic literature has focused on the altered expression of ACE2 in children and adults affected by SARS-CoV-2 as the primary outcome.

## 4. Discussion

Children have the same risk of infection with SARS-CoV-2 as the general population, with the most severe cases found in newborns [45,47]. However, disease severity is frequently lower in children than in adults, with a benign or moderate course, in the absence of risk factors or concomitant diseases [48,49].

### 4.1. Innate Immunity to SARS-CoV-2 in Children

Epithelial cells, lymphoid cells, alveolar macrophages, and dendritic cells represent the first barrier that respiratory viruses (rhinoviruses, respiratory syncytial virus, influenza, and coronaviruses) encounter, representing the cornerstone of the innate immune response from the nasopharynx to the alveolar epithelium. The purpose of this innate defensive barrier in the early stages of infection is to promote the elimination of the virus, inhibit viral replication, induce tissue repair, and initiate a subsequent specific adaptive immune response [50].

The cells involved in this response in children are mainly alveolar macrophages (the first to come into contact with respiratory viruses and present in greater numbers in children), which, together with neutrophils and epithelial cells, produce antiviral proteins triggered by viral presence through sensors that generate a cascade of intracellular and intercellular signals responsible for the activation of immunity and the release of antiviral proteins.

Specifically, both the innate and adaptive immune systems are involved in the first phase of the physiological response to SARS-CoV-2. After ACE2-mediated entry into cells, the viral patterns are recognized by Toll-like receptors (TLRs) expressed by innate immune cells (TLR-7) [51]. A recent study conducted by Yin et al. identified MDA5 (melanoma differentiation-associated protein 5) and LGP2 (laboratory of genetics and physiology gene 2) as the primary sensors for SARS-CoV-2 infection, regulating the induction of IFN in response to infection [52]. Activation of TLRs leads to the production of pro-inflammatory cytokines and IFN (IFN-1 and IFN-III) [53], which promote antiviral effects, improving phagocytosis and the chemotaxis and activation of natural killer cells (NK), initiating the adaptive humoral response. Furthermore, following phagocytosis, viral proteins are internalized in macrophages and presented to class I and II major histocompatibility complexes, activating the adaptive immune response [54].

This response is more effective in children and not as effective in infants; moreover, the response would be favored, compared to adults, by the presence of cleaner lung conditions, a more optimal state of health or genetic background, such as the Human Leukocyte Antigen (HLA) system that ensures an immune response with a specific anti-SARS-CoV-2 immune locus. There is still no definitive evidence that the adaptive immune response, which occurs later, against SARS-CoV-2 is more active in children. The response is modulated by the CD4 + helper T lymphocytes, which stimulate B lymphocytes during the antibody response, and Treg lymphocytes, which are prevalent in respiratory tissues in children [39,55].

### 4.2. Role of the Differential Expression of ACE2 in Children in the Etiopathogenesis of COVID-19

SARS-CoV-2 uses ACE2 as a receptor, TMPRSS2 and cathepsin L (CTSL) as a promoter, which facilitate access to cells [56,57] (Figure 2). Also facilitating access to cells is furin, a ubiquitously expressed protease with the ability to cleave some envelope glycoproteins from a wide range of viruses, facilitating viral fusion with cell membranes. The furin cleavage site is between subunits 1 and 2 (S1/S2) and is in a slightly different position from the activation site of TMPRSS2 and cathepisin L [58].

ACE2 appears to be expressed mostly on the apical surface of the well-differentiated cells lining lung alveoli, where it causes the greatest damage [59]. In particular, according to a recent study conducted by Ortiz Bezara et al., the apical cell surfaces of a small subgroup of alveolar type II cells [40], are mainly involved with SOX9 positive lung progenitor cells (pluripotent stem cells also present in the lungs of children) [45]. In addition, strong expression was also demonstrated in the mouth and tongue, implicating the oral cavity as a route of infection [60].

An important role in the differentiation of response to the entry of the virus into the cell could be played by the difference in the positioning of ACE2, which should be superficial and apical, to favor its binding with the viral protein S spike [61].

In children, this expression could be different, as some studies report that the number and function of ACE receptors are lower in children than in adults, and the expression increases from 40 years of age, with a peak between 60–80 [54,62]. In addition, lower ACE2 and TMPRSS2 expression in nasal and bronchial epithelial cells in children compared to adult COVID-19 patients was confirmed by Sharif-Askari et al. [41] and by Bunyavanich et al. [35] in nasal epithelium ACE2. Interestingly, in a study conducted by Somekh et al. [34], out of 20 families who contracted COVID-19, olfactory and gustatory sensations were gradually but significantly less impaired among children (aged 5 and 10 years) than in adults. Furthermore, Somekh et al. correlated sensory impairment with ACE2 expression in the corresponding age groups with sensory impairment scores and relative ACE2 expression using the data from the literature (Pearson's correlation coefficient = 0.95, $p = 0.05$) [34].

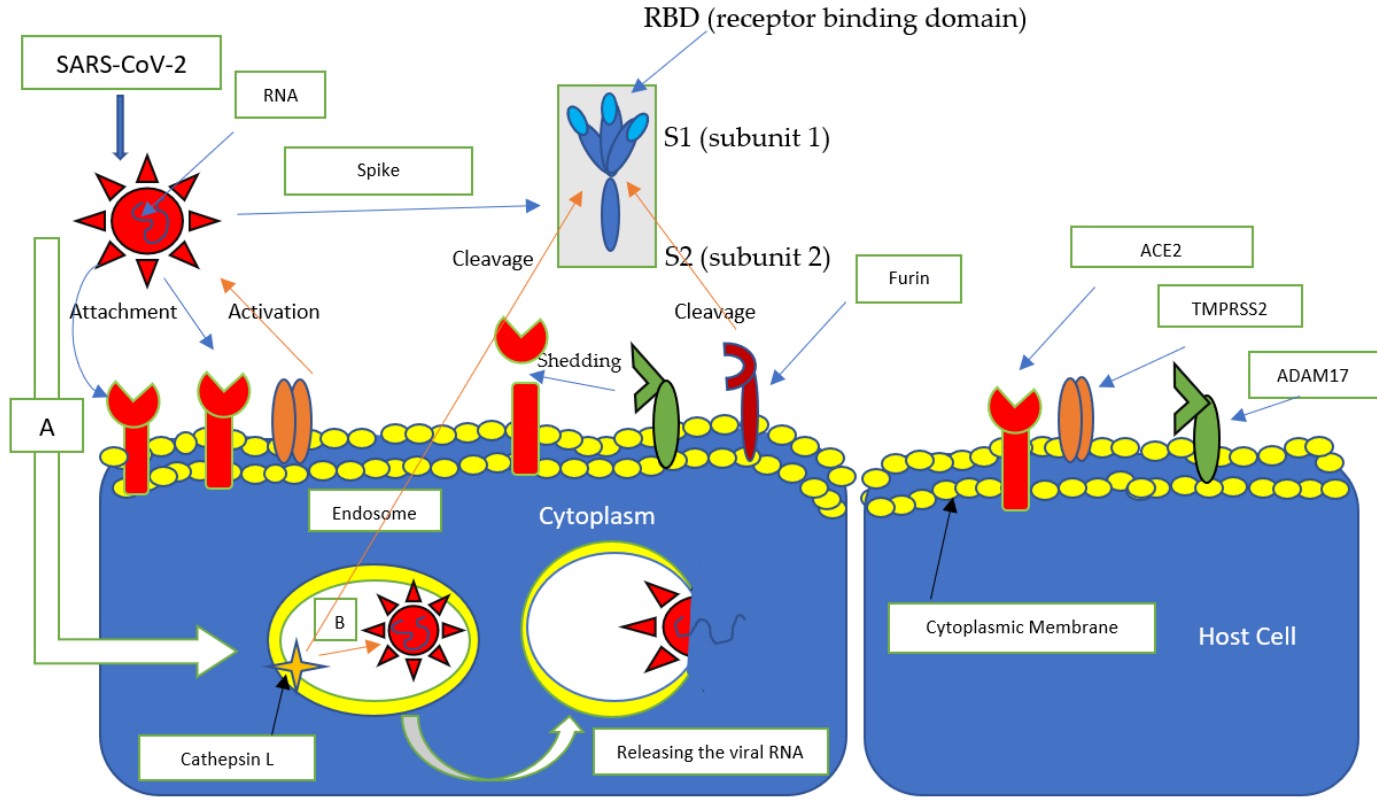

**Figure 2.** The primary receptors involved in the entry of SARS-CoV-2. SARS-CoV-2 binds to the ACE2 receptor via the spike protein; specifically, ACE2 binds to the S1 subunit at the RBD (Receptor Binding Domain) level, TMPRSS promotes binding by interacting with the spike protein, while Furin detaches the bond between the S1 and S2 subunits, favoring the fusion of the virus capsid with the cytoplasmic membrane. ADAM17 is responsible for the shedding of the ACE2 receptor. (**A**) The virus penetrates cells through endocytosis, after binding with the ACE2 receptor, (**B**) forming an endosome where cathepsin L cleavage of the spike protein releases viral RNA.

The data presented to date suggest a positive relationship between the expression of ACE2 and COVID-19 susceptibility; however, the scientific evidence indicates that the downregulation of ACE2 leads to a worsening of inflammatory foci due to overexpression of angiotensin 2 in the RAAS system [63,64]. Furthermore, in vitro reports suggest that SARS-CoV-2 downregulates ACE2 after penetrating cells [64].

In addition, circulating soluble ACE2 enzyme, which tends to bind to the spike protein of SARS-CoV-2, could play a role in reducing the possibility of the virus to bind to the membrane-bound ACE2 receptor. Therefore, the plasma profile of ACE2 could play a role in providing greater resistance in the pediatric population, since plasmatic ACE2 is more present in children [65]. According to a study by Bénéteau-Burnat et al., in children aged 6 months to 17 years, ACE2 levels were 13–100 U/liter compared to 9–67 U/liter in adults [65,66], but these data do not agree with a study conducted by Sharif-Askari et al., who found no difference in plasmatic expression levels of ACE2 and TMPRSS2 between children and adults [41].

However, in a letter to the editor in the European Respiratory Journal by Porter, there is still doubt about the role of ACE2 expression in the etiopathology and prognosis of COVID-19 [67,68].

*4.3. The Role of Infections and Past Infections in the Etiopathogenesis of COVID-19 in Children*

Children have high infection rates in the first years of life that subsequently tend to decrease; among these, common human coronavirus (CovH) can infect people of all ages

by generating only relative immunity, while a high incidence of SARS-CoV-2 infection in adults would indicate poor cross-immunity with previous CovH infections.

On the other hand, children and young adults have increased antibodies against CovH with cross-reactivity with SARS-CoV [69].

Coinfection in children has been demonstrated for CovH in about two-thirds of cases; coinfection seems to have antagonistic effects, as infection by rhinoviruses of the nasopharyngeal tract can block the growth of other viruses, with the virus with the highest viral load becoming predominant in upper respiratory tract infections.

Therefore, exposure in winter and spring [70], in which the immune system is more active against respiratory viruses, is also more active against SARS-CoV-2 infection. Furthermore, CovH (NL63), associated with the common cold, produces a downregulation of ACE2 [71], a reduction that could partly explain why children with CovH colds are hospitalized less frequently than adults [72].

In a study conducted in 2020 in the USA by Schneider et al. in children with flu symptoms, it was found that 22.2% of children infected with SARS-CoV-2 and 37.1% of children not infected with SARS-CoV-2 had one or more non-SARS-CoV-2 pathogens; in this study, infection with other viruses did not rule out COVID-19 infection [73]. These data do not disagree with data reported by Wu et al., in which half of the children infected with SARS-CoV-2 had coinfection with other common respiratory pathogens [74].

Li et al. reported that coinfection was relatively common in children with COVID-19, with nearly one-third of children having a coinfection. The most frequent coinfection was *Mycoplasma pneumoniae* (25%), followed by viruses (7%) and other bacteria (5%); coinfection did not cause worsening of clinical manifestations [75].

From a very early study conducted by Wölfel et al. in Germany, it was found that there was no viral coinfection in adults with SARS-CoV-2 [76]. In partial agreement with these data, Kim et al., in April 2020, concluded that the presence of a non-SARS-CoV-2 pathogen might not guarantee that a patient does not also have SARS-CoV-2 [77]. In adults, the main viruses involved in coinfections with SARS-CoV-2 are rhinoviruses, IV, and CovH [78,79], while *Mycoplasma pneumonia* appears to be among the most commonly coinfected microorganisms in patients affected by COVID-19 [80].

Stowe et al. concluded in a study performed in England that the risk of testing positive for SARS-CoV-2 was 58% lower among influenza-positive cases, suggesting possible pathogenic competition between the two viruses, but also reported that coinfected patients had a 2.27-fold greater risk of death compared to SARS-CoV-2-borne infection alone, suggesting possible synergistic effects in coinfected individuals. Thus, these latest data suggest a competitive effect during the early stages of infection, however later coinfection certainly represents a risk factor for death [81].

In support of this hypothesis is a study conducted by Schweitzer et al., in which human small airway epithelial cells of healthy patients were infected with the IV A virus, leading to an amplification in the expression of ACE2, sheddase, ADAM17 (a disintegrin and metalloproteinase 17), (TNF)-alpha converting enzyme (TACE), and TMPRSS2, which are involved in the penetration of SARS-CoV-2 into cells [42].

A limitation of this systematic review is the lack of information present in the studies carried out. In fact, most of the studies that investigated the differences in ACE2 expression in tissues of lung origin in children were on patients not infected with SARS-CoV-2. Most of the data originated from previously acquired databases (asthmatic children), and in some cases, the expression of ACE2 was evaluated, but with a limited number of samples.

## 5. Conclusions

The scientific literature from the first months of the pandemic agrees that children develop milder COVID-19 disease than adults. The milder symptomatologic frame could be attributed to innate immunity or previous CovH virus infections, while it is still not entirely clear how the differential expression of ACE2 in children could cause a milder disease.

**Author Contributions:** Conceptualization, M.D., D.S., A.P.C.; L.L., V.C. and G.T.; Methodology, M.D. and A.D.; Software, M.D.; Data analysis, M.D., A.D., G.A.C. and D.S.; Visualization, M.D.; Supervision and project administration, L.L.M.; Writing, M.D.; Reviewing and editing, M.D. and C.A. All authors have read and agreed to the published version of the manuscript.

**Funding:** This research received no external funding.

**Institutional Review Board Statement:** Not applicable.

**Informed Consent Statement:** Not applicable.

**Data Availability Statement:** Data sharing not applicable.

**Conflicts of Interest:** The authors declare no conflict of interest.

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
