# Peer review of "Innate Immunity in Children and the Role of ACE2 Expression in SARS-CoV-2 Infection"

_pediatrrep, doi:10.3390/pediatric13030045_

Round 1

Reviewer 1 Report

Thank you for correction and editing this article as per comments. 

Author Response

Thank you for helping us improve the manuscript

Best regards 

Mario Dioguardi

Reviewer 2 Report

This is a re-submitted manuscript by Dioguardi et al., which reviewed and discussed the cause of COVID-19 symptoms that is milder in children than in adults. Since the authors respond to the first review and have addressed my concerns, my comments are minor, which are listed below to improve the quality of the manuscript.

Suggestions/Comments

  1. After the first appearance of the abbreviation, the abbreviation should always be used in the rest of the manuscript, such as IFNs (Line 241) and TMPRSS2 (Line 257), Influenza Virus (Line 347).
  2. Line 41, either Angiotensin-Converting Enzyme 2 or ACE-2 should be removed.
  3. In figure 1, although the author explained the method to identify 18 studies, the authors should also describe in the Flow chart. Also, the number of records excluded from 556 is not 301 and must be 204 in the Flow chart.
  4. In Figure 2, TMPRSS and Sars-CoV-2 should be TMPRSS2 and TMPRSS2, and the author should explain the role of furin in the manuscript. Also, it is hard to understand that Cathepsin L cleaves the Spike protein and release viral RNA in the figure.

Author Response

Reviewer 2

This is a re-submitted manuscript by Dioguardi et al., which reviewed and discussed the cause of COVID-19 symptoms that is milder in children than in adults. Since the authors respond to the first review and have addressed my concerns, my comments are minor, which are listed below to improve the quality of the manuscript.

Suggestions/Comments

  1. After the first appearance of the abbreviation, the abbreviation should always be used in the rest of the manuscript, such as IFNs (Line 241) and TMPRSS2 (Line 257), Influenza Virus (Line 347).
  2. Line 41, either Angiotensin-Converting Enzyme 2 or ACE-2 should be removed.
  3. In figure 1, although the author explained the method to identify 18 studies, the authors should also describe in the Flow chart. Also, the number of records excluded from 556 is not 301 and must be 204 in the Flow chart.
  4. In Figure 2, TMPRSS and Sars-CoV-2 should be TMPRSS2 and TMPRSS2, and the author should explain the role of furin in the manuscript. Also, it is hard to understand that Cathepsin L cleaves the Spike protein and release viral RNA in the figure.

Answer:

thanks for the comments and suggestions, they were very helpful in improving the manuscript

  • Checked and modified as required
  • Removed Angiotensin-Converting Enzyme 2
  • modified figure 1 (flow chart)

  • modified figure 2. Added information on Furin in the manuscript: And furin which is a ubiquitously expressed protease and has the ability to cleave some envelope glycoproteins of a wide range of viruses, facilitating viral fusion with cell membranes, the furin cleavage site is between subunits 1 and 2 (S1 \ S2) and is in a slightly different position from the activation site of TMPRSS2 and cathepisin L.

This manuscript is a resubmission of an earlier submission. The following is a list of the peer review reports and author responses from that submission.

Round 1

Reviewer 1 Report

The review article by Dioguardi et al. entitled "Innate immunity in children and the role of ACE2 expression 2 in SARS-CoV-2 infection" explore and reviewed about children develop milder Covid-19 disease than adults, the reasons for a milder course of the disease were attributed to several factors: innate immunity, the difference in ACE2 receptor expression and previous infections as well as other common Coronaviruses.

Significance of the study: The literature review aims to summarize aspects related to innate immunity by focusing on the role of ACE2 expression and the role of viral infections contracted in children in modulating the antibody response to SARS-29 CoV-2 infection.

However, the author's subsequent remarks need to address in revision to clarify and qualify their investigation's impact and meaning.

  1. Poorly edited review article.
  2. Please be consistent about naming ace2/ACE2 SARS-Cov2 full form etc..
  3. The introduction needs to rewrite by focusing on why you carry out this study.
  4. Please correct the reference numbering through manuscripts.
  5. What reader will learn from your review?
  6. Make two 2-3 figures showing about ACE2 and COVID-19 etc.
  7. References arrange according to the journal format.

Reviewer 2 Report

Dear Authors,

I read with interest your manuscript. Please find attached my comments below

Introduction: Authors should present the potential differences and recent original studies that address these differences between pediatric and adult patients. General information regarding COVID-19 should be avoided since they are widely known now.

Methods: Authors should present in more clarity the eligibility and exclusion criteria for their study.

Results: Authors should describe in more detail the results of their literature review and add more data in their results section

Discussion: After revision of the results section authors should discuss point by point their findings and try to compare their results with other studies. 

Reviewer 3 Report

 In the manuscript entitled "Innate Immunity in children and the role of ACE2 expression in SARS-CoV-2 infection." by Dioguardi et al., the authors review and discuss the COVID-19 in children by focusing on innate immunity and ACE2 expression. The manuscript is original, and the review area is interesting; however, it is not well written and difficult to follow with several misspellings and grammatical mistakes. I would suggest that the manuscript needs English proofreading by a specialized and native speaker. Also, Table 1 and Figure 1 are partly redundant, and one is probably not necessary. Besides, the role of ACE2 in the children in Table 2, one of the main parts of this review, is not informative and valuable since the author extracts from the original manuscript. Therefore, I would suggest that the authors mention or summarize these references in table 2 at least once in the manuscript.

Some other concerns are listed below to improve the quality of the manuscript.

Suggestions/Comments:

  1. Lacking spell out of the abbreviation such as COVID-19, SARS-CoV-2, MERS, ACE2, and TLR, etc. Abbreviations should be spelled out on the first appearance.
  2. Line 59, I think references #22 did not mention about animal studies on SARS-CoV.
  3. Please explain "electronic database i" in Line 103.
  4. The authors suggested that the number of qualitative analysis is 17 in Line 127. Is this correct, not 18?
  5. Line 142, the reference [42] should be removed from Schweitzer et al. 2021[41], [42].
  6. Line 152 and Line 208, Covi-19 or Covd-19 should be changed to COVID-19.
  7. In section 4.1, the authors should provide a reference that reported that SARS-CoV-2 was detected by TLR-7. However, recently, Yin et al. reported that MDA5 and LGP2 are the major sensors for SARS-CoV-2 infection (Cell Rep. 2021 Jan 12;34(2):108628. doi: 10.1016/j.celrep.2020.108628.). The author needs to mention these innate immune sensors in the manuscript.
  8. Line 191, coronavirus should be removed from the manuscript.
  9. Line 215, the authors should cite reference ï¼»31ï¼½ at the end of the sentence.